

# Novel statistically equivalent signature-based hybrid feature selection and ensemble deep learning LSTM and GRU for chronic kidney disease classification

Yogesh N[1,2], Purohit Shrinivasacharya[1,2] and Nagaraj Naik[3]

[1] Siddaganga Institute of Technology, Tumkuru, Karanataka, India
[2] Visvesveraya Technological University, Belagavi, India
[3] Computer Science & Engineering, Manipal Institute of Technology, Manipal Academy of Higher Education (MAHE), Manipal, Karanataka, India

Corresponding author
Nagaraj Naik,
nagaraj.naik@manipal.edu,
nagaraj21.naik@gmail.com

## ABSTRACT

Chronic kidney disease (CKD) involves numerous variables, but only a few significantly impact the classification task. The statistically equivalent signature (SES) method, inspired by constraint-based learning of Bayesian networks, is employed to identify essential features in CKD. Unlike conventional feature selection methods, which typically focus on a single set of features with the highest predictive potential, the SES method can identify multiple predictive feature subsets with similar performance. However, most feature selection (FS) classifiers perform suboptimally with strongly correlated data. The FS approach faces challenges in identifying crucial features and selecting the most effective classifier, particularly in high-dimensional data. This study proposes using the Least Absolute Shrinkage and Selection Operator (LASSO) in conjunction with the SES method for feature selection in CKD identification. Following this, an ensemble deep-learning model combining long short-term memory (LSTM) and gated recurrent unit (GRU) networks is proposed for CKD classification. The features selected by the hybrid feature selection method are fed into the ensemble deep-learning model. The model's performance is evaluated using accuracy, precision, recall, and F1 score metrics. The experimental results are compared with individual classifiers, including decision tree (DT), Random Forest (RF), logistic regression (LR), and support vector machine (SVM). The findings indicate a 2% improvement in classification accuracy when using the proposed hybrid feature selection method combined with the LSTM and GRU ensemble deep-learning model. Further analysis reveals that certain features, such as HEMO, POT, bacteria, and coronary artery disease, contribute minimally to the classification task. Future research could explore additional feature selection methods, including dynamic feature selection that adapts to evolving datasets and incorporates clinical knowledge to enhance CKD classification accuracy further.

# INTRODUCTION

Chronic kidney disease (CKD) has emerged as a primary focus in contemporary medical research, according to recent studies (*Shakhshir et al., 2023*; *Islam, Majumder & Hussein, 2023*). CKD affects millions of people worldwide and is linked to higher chances of getting sick, dying, and spending more money on healthcare (*Levey et al., 2007*; *Lv & Zhang, 2019*). Complications and disease progression can be minimal with early detection and personalized therapies. Therefore, it is important to identify relevant variables in CKD that improves the classifier performance.

CKD is characterized by 24 distinct features. Proper evaluation of each feature is essential to improve the accuracy of classification tasks. However, some features have a minimal impact on classification performance and should be removed to decrease computational demands and enhance model efficiency. Most existing studies have concentrated on methods for selecting individual features for CKD classification. Yet, it is vital to identify and discard irrelevant features that do not significantly aid the classification process, as this also helps reduce computational time. Hence, employing a hybrid feature selection strategy is necessary to identify the most crucial features of CKD. Removing redundant features can reduce noise and complexity in the model. The ultimate goal of this paper is to create a state-of-the-art CKD classification model with high accuracy. This would help doctors better understand all the variables that play a role in diagnosing chronic kidney disease. According to *Tannor et al. (2019)* and *Bai et al. (2022)*, the dataset used for this study includes critical clinical factors such as gender, age, blood pressure, laboratory results, and medical history.

Feature selection is an important task in CKD classification. Typically, it identifies relevant tasks' features, such as regression and classification, maintaining only the variables with the highest predictive power. Feature selection aims to improve classification performance, reduce computational cost, and eliminate unnecessary or redundant features. In order to address the challenge posed by the numerous features in diagnosing CKD, it is crucial to consider advanced feature selection techniques to identify the most significant variables that contribute to the classification process. This can help streamline the diagnostic process and improve the accuracy of CKD classification models (*Sawhney et al., 2023*; *Harimoorthy & Thangavelu, 2021*). Selecting relevant features from the entire feature space can enhance model performance by reducing dimensionality and removing noise. Few studies have investigated effective techniques adapted to the classification task, even if feature selection is crucial (*Fahimifar et al., 2023*; *Habibi et al., 2023*). Therefore, a hybrid feature selection methodology is developed in this study by combining the statistically equivalent signature (SES) and Least Absolute Shrinkage and Selection Operator (LASSO) feature selection methods.

Machine learning techniques have shown promising results in medical diagnosis and decision-making (*Ebiaredoh-Mienye et al., 2022*; *Ferguson et al., 2022*; *Lee et al., 2022*). Neural networks like convolutional neural networks (CNN) and recurrent neural networks (RNN) can automatically learn and extract complex patterns from large datasets. This helps in many medical areas, such as image analysis and disease diagnosis for accurate

classification (*Cui & Liu, 2019*; *Shi et al., 2019*). Long short-term memory (LSTM) networks are helpful in classification tasks because they can effectively model and capture dependencies in sequential data (*Hwang et al., 2019*; *Garcia et al., 2020*). Unlike traditional feedforward neural networks, LSTM can remember information for long periods, making them well-suited for analyzing and classifying sequential data such as time series, text, and speech (*Ju & Liu, 2021*). Their ability to retain and utilize information over extended periods allows them to capture complex patterns and relationships within the input data. This is particularly beneficial for classification tasks where long-range dependencies are important. Additionally, LSTMs can handle variable-length sequences, making them versatile for a wide range of classification tasks. Gated recurrent unit (GRU) networks are helpful in classification tasks because they can effectively capture and model dependencies in sequential data (*Rana, 2016*; *Zulqarnain et al., 2021*). The gating mechanism enables them to retain and update information selectively, capturing long-range dependencies and complex patterns within the input data. Ensemble learning is a powerful technique in machine learning. It improves classification performance by combining multiple classifiers. Ensemble learning's success relies on the individual classifiers' effectiveness and ability to generalize.

The following are the three main types of ensemble learning:

- Bagging: Bagging, a technique proposed by *Breiman (1996)*, aims to improve classification accuracy by combining the predictions of models trained on deliberately constructed training sets. A notable implementation of bagging is the random forest algorithm, which utilizes an ensemble of random decision trees to achieve robust classification results.
- Boosting: Boosting, an ensemble meta-algorithm, is a powerful technique for generating classifiers by combining multiple weak learners. This method involves iteratively training new models, with a focus on samples misclassified by earlier models, to construct an ensemble gradually. Despite developing newer algorithms that may yield better results, Adaboost remains the most widely used boosting implementation.
- Bucket of models: In the bucket of models ensemble learning technique, a model selection algorithm is employed to identify the most suitable model from a collection for addressing diverse challenges. Cross-validation is the prevailing method utilized for model selection in this context.

The research gap in the existing literature on CKD prediction techniques is limited by exploring hybrid feature selection methods. While numerous studies emphasize single-feature selection techniques, these approaches often fail to consider the interaction between features and their combined influence on model performance. Current methods frequently struggle to effectively eliminate redundant or irrelevant features, which can result in increased computational complexity, potential overfitting, and decreased model accuracy. The proposed hybrid feature selection method is necessary because it enhances the classification task by identifying the most relevant features.

Traditional methods, such as machine learning, may struggle to fully capture the inherent complexity of CKD. Therefore, ensemble deep-learning techniques are required to identify relevant patterns in the data.

The contribution of this work is as follows:

- Proposed hybrid feature selection method using LASSO feature selection and SES approach for identifying features related to CKD
- Proposed a combined LSTM and GRU ensemble deep learning model to achieve higher accuracy for the CKD classification task.

The rest of this study is structured as follows: 'Related Work' reviews the literature on CKD classification, feature selection methods, and deep learning models in healthcare. 'Proposed Work' discusses the methodology. 'Experimental Results' describes the experimental findings and results. Finally, 'Conclusion' summarizes the contributions and significance of the proposed work.

## RELATED WORK

*Saif, Sarhan & Elshennawy (2023)* introduce three predictive models for the early detection of Chronic Kidney Disease (CKD): convolutional neural networks (CNNs), LSTM networks, and a deep ensemble model. The deep ensemble model, which combines CNN, LSTM, and Bidirectional LSTM (BLSTM) classifiers using majority voting, demonstrates superior performance. It achieved accuracy rates of 0.993 for 6-month predictions and 0.992 for 12-month predictions, outperforming the individual models.

The high-performing CKD prediction framework was developed using deep learning and ensemble methods, including CNN, LSTM, and LSTM-BLSTM, to forecast CKD occurrences 6 to 12 months in advance (*Saif, Sarhan & Elshennawy, 2024*). The proposed framework addresses data imbalance and optimization challenges, achieving accuracies of 98% and 97% for 6 and 12 months, respectively, and significantly surpasses previous methods in early disease prediction.

*Ghosh et al. (2020)* proposed four reliable methods to classify CKD using a support vector machine (SVM), AdaBoost, linear discriminant analysis, and gradient boosting. These methods are deployed on the University of California Irvine (UCI) machine learning repository dataset, available online. The study concludes that AdaBoost classifiers have the highest predicted accuracy.

*Aljaaf et al. (2018)* presented the performance of multiple machine-learning techniques for CKD early prediction. Although much research has already been done on this topic and using predictive analytics to back up our methods by looking at how different data parameters relate to each other and to the attribute we are trying to forecast. Using predictive analytics, select the best set of inputs for our machine-learning prediction models. In order to predict CKD, this study begins with 24 characteristics in addition to the class property. Four machine learning-based classifiers were compared in a supervised learning environment, and multilayer perceptron outperformed.

Preemptively identifying chronic renal disease using data mining and machine learning approaches is considered to reduce the number of patients and the associated treatment

costs (*Alassaf et al., 2018*). The work considered data from the King Fahd University Hospital (KFUH) database in Khobar, Saudi Arabia. The experimental results demonstrate the superior accuracy of artificial neural network (ANN) and SVM over K-nearest neighbor (KNN).

*Qin et al. (2019)* proposed using a machine-learning approach to classify CKD. Multiple missing values plague the CKD data set retrieved from the machine learning repository at UCI. KNN imputation, which chooses several total samples with the most similar measurements to process the missing data for each partial sample, was used to fill in the missing values. Six machine methods were considered, namely logistic regression (LR), Random Forest (RF), support vector machine (SVM), KNN, naive Bayes classifier, and feed-forward neural network, which were employed. RF performed the best of these machine learning models.

Five hundred and fifty-one individuals with proteinuria had their clinical and blood chemistry data collected (*Xiao et al., 2019*). Non-urine clinical indicators were employed to predict the 24-hour urinary protein outcome response, and they included 13 blood-derived tests and five demographic parameters. Nine types of predictive models were developed and compared, including LR, Elastic Net, LASSO regression, ridge regression, SVM, RF, XGBoost, neural network, and KNN. Each model was tested for its area under the receiver operating characteristic (AU-ROC), recall, and accuracy.

*Ma et al. (2020)* proposed a Heterogeneous Modified Artificial Neural Network (HMANN) for the early detection, segmentation, and diagnosis of chronic renal failure. The proposed technique utilizes an ultrasound image, with the segmentation of the region of renal interest being referred to as a preprocessing step. The proposed HMANN technique reduces segmentation time and improves accuracy in kidney segmentation. *Mezzatesta et al. (2019)* considered SVM learning method to classify the CKD. The radial basis functions (RBF) kernel and GridSearch methods were considered to fine-tune the hyper-parameters. *Saif, Sarhan & Elshennawy (2024)* proposed a deep learning and ensemble framework to predict chronic kidney disease (CKD). The framework addresses key challenges like data imbalance and feature selection, significantly improving early CKD prediction. *Nramban Kannan et al. (2024)* proposed a reinforcement learning-based artificial neural network (RL-ANN) model for CKD classification.

An extensive literature review has been carried out in the field of chronic kidney diseases. The literature review summary is described in Table 1. Most of the work considered a single classifier for CKD classification. Hybrid feature selection and ensemble classifiers are popular methods to improve classification accuracy when data is nonlinear. Therefore, we proposed a hybrid selection-based ensemble classifier for CKD.

## PROPOSED WORK

There are 24 features in CKD described in Table 2. Assessing each feature of CKD is essential to improve the classification task. Some features contribute little to accurate classification tasks. Hence, it is necessary to remove the irrelevant feature from the list. Most of the work has been done on single-feature selection methods for CKD classification.

**Table 1  Summary of related work.**

| Author | Method | Merit | Demerit | Feature selection |
|---|---|---|---|---|
| *Arif, Mukheimer & Asif (2023)* | KNN (K-Nearest Neighbour) and Gaussian NB | Boruta effectively capture nonlinear relationships within the data | A single feature selection method like Boruta may not capture a dataset's complex feature relationships and interactions. | Yes |
| *Swain et al. (2023)* | SVM and Random Forest (RF) | RF ensemble of decision trees reduces overfitting, making it a more versatile and reliable CKD classification model. | RF with many trees can be extremely computationally and resource-intensive. | No |
| *Ghosh et al. (2020)* | SVM and AdaBoost | SVM are good for CKD classification because they can handle nonlinear correlations. Adaboost improves accuracy by merging weak classifiers | CKD classification is sensitive to parameter adjustment and may struggle with massive datasets. Adaboost may be subject to outliers and noisy data. | No |
| *Aljaaf et al. (2018)* | SVM and Multilayer Perceptron | It is effectiveness in handling nonlinear relationships and robustness in high-dimensional feature spaces | Complex multilayer perceptron (MLP) neural networks may require careful tuning and overfitting, especially with little data. | No |
| *Alassaf et al. (2018)* | ANN and SVM | The experimental results demonstrate the superior accuracy of ANN and SVM over the KNN method | ANN for CKD may have a higher computational cost, which makes them unsuitable for situations where computational resources are restricted. | No |

However, it is essential to describe the unimportant feature that does not contribute to the classification task and increases the computational time in the model. A hybrid feature selection approach is needed to identify the critical features of CKD. Due to noise and model complexity, redundant features in a dataset might reduce accuracy. Overfitting can occur when a dataset has similar or duplicate features, making it difficult for the model to find patterns and relationships. The overall proposed work are described in Fig. 1.

Figure 2 shows the proposed framework. The work is divided into two phases. Phase 1 involves assessing each feature of CKD to check how features are relevant for classification tasks. Therefore, hybrid feature selection is considered to identify the essential features of CKD. The hybrid feature selection method combines the SES and LASSO feature selection methods.

## SES feature selection

In the proposed work, the SES method is considered to identify the important variable in CKD. The SES method is a constraint-based feature selection technique grounded in causal analysis theory. This approach identifies the optimal set of predictors for a target variable, known as the Markov blanket (MB), within a Bayesian network (BN) under specific assumptions. Bayesian networks offer compact representations of multivariate distributions using a direct acyclic graph (DAG) and precise parameterization, where nodes represent random variables and edges signify conditional relationships. In conditional relationships, if an edge connects two nodes, their variables exhibit association among all other variables. The SES method, as hypothetically implemented here using the Gini Index, assesses the

**Table 2 CKD feature description.**

| Feature No | Name | Description |
|---|---|---|
| 1 | Age | Patient age |
| 2 | BP | Blood pressure |
| 3 | SG | Specific gravity |
| 4 | AL | Albumin |
| 5 | SU | Sugar |
| 6 | RBC | Red blood cells |
| 7 | PC | Pus cell |
| 8 | PCC | Pus cell clumps |
| 9 | BA | Bacteria |
| 10 | BGR | Blood glucose random |
| 11 | BU | Blood urea |
| 12 | SC | Serum creatinine |
| 13 | SOD | Sodium |
| 14 | POT | Potassium |
| 15 | HEMO | Hemoglobin |
| 16 | PCV | Packed cell volume |
| 17 | WC | White blood cell count |
| 18 | RC | Red blood cell count |
| 19 | HTN | Hypertension |
| 20 | DM | Diabetes mellitus |
| 21 | CAD | Coronary artery disease |
| 22 | APPET | Appetite |
| 23 | PE | Pedal edema |
| 24 | ANE | Anemia |
| 25 | CLASS | Diagnosis ckd, notckd |

significance of each feature by examining its statistical impact on the model. Traditionally, the Gini Index is employed in decision trees to quantify node impurity, but in this scenario, it is utilized to gauge the relevance of individual attributes. Features are then ranked based on their importance scores, with those exceeding a specified threshold identified as significant.

In Algorithm 1, presented in pseudocode, SES is outlined. This method takes as input a dataset $D\_set$, a target variable $Tar$, and two hyperparameters: a threshold for discerning conditional independence $thres$ and an upper limit $max$ on variables within a conditional set. These parameters constrain the algorithm's computational complexity and resource requirements. The procedure generates a set $K$ of variables, organized into queues $Q$ with $i = 1$ to $N$ and each queue containing equivalent variables.

During initialization, an empty set $Set\_s$ of selected variables is created, encompassing all variables $v$ and added in $Set\_s$, and $L \leftarrow v$ represents the list of variables, and each variable is equivalent only to $Q_i \leftarrow i$. The algorithm runs a loop, and the aim is to include the variable most correlated with $Tar$ based on any subset of selected variables and exclude

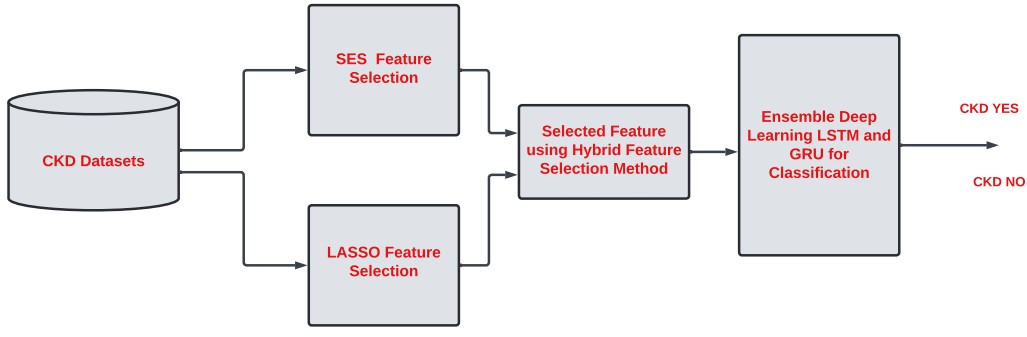

**Figure 1  Overall work.**

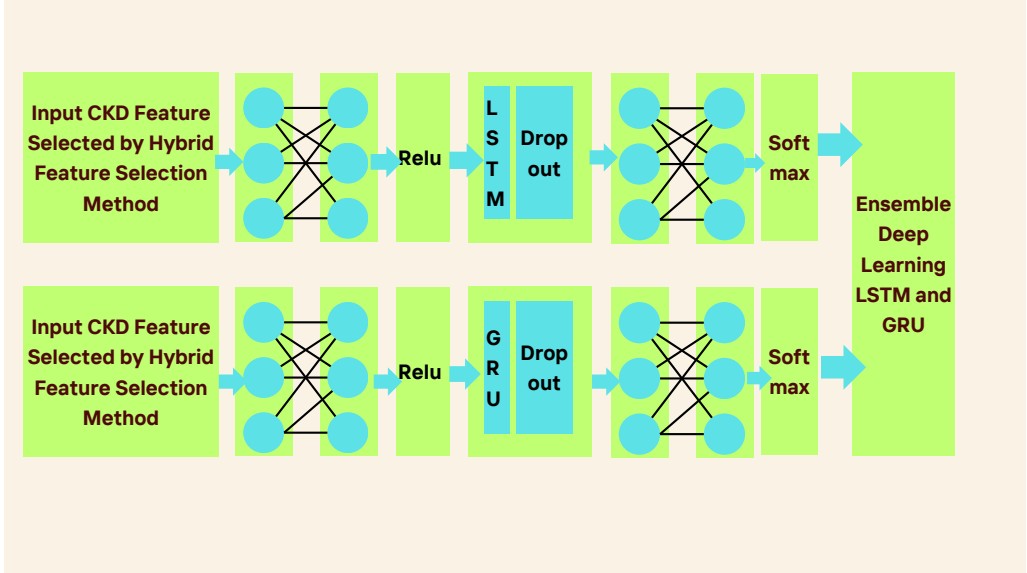

**Figure 2  Ensemble deep learning LSTM and GRU for CKD classification.**

any variable $v$ not correlated with $Tar$ given any subset $sub\_set$ of other variables in $Set\_s$. Excluded variables $v$ are ineligible for addition to $Set\_s$.

Before eliminating $v$ from $Set\_s$, the method searches for a variable $p$ in $sub\_set$ that matches $v$ by validating when $sub\_set \leftarrow sub\_set \cup \{v\} \setminus \{p\}$.

## LASSO feature selection

LASSO, a linear model utilizing L1 regularization, reduces the coefficients of less important features to zero, thereby conducting feature selection. The significance of each feature is reflected by the absolute value of its coefficient. Features with non-zero coefficients are deemed important and are included in the final model. LASSO feature selection enhances model interpretation and prediction accuracy. It uses ridge regression and subset selection simultaneously and selects one feature correlation while minimizing the others to zero. All regression methods aim to find coefficients for existing features, ensuring that applying

these coefficients to each sample of data yields the defined outcome. The LASSO approach truncates coefficients to zero by changing them by a constant value of λ. Reduces residual sum of squares, resulting in a reduced absolute value of coefficients. The following is a definition of LASSO in its formal form, which is defined in the Eq. (1). The LASSO formula aims to discover the optimal coefficients $\beta$ that balance model complexity and predictive accuracy by minimizing the sum of the L1 regularization penalty and the squared loss term.

$$\widehat{\beta} = arg_\beta min \sum_{i=1}^{N} \left( y_i - \beta_0 \sum_{j=1}^{p} v_{ij}\beta_j \right)^2 \tag{1}$$

---

**Algorithm 1** Hybrid SES feature selection algorithm

---

1: Input CKD Datasets.
2: $D\_set$ represents datasets
3: $v$ independent variables
4: $Tar$ represents dataset target variables
5: $thres$ represents threshold
6: A collection of data on n independent predicting variables i.
7: The variable t is the target.
8: Indicator of threshold th variable
9: Max set is representing variable $max$
10: A set $max$ of $N$ variables $Q_i = 1 to N$ can be used to generate a signature by selecting only one variable from each set.
11: List of variables $L \leftarrow v$
12: Variables selected.
13: Set $\_s \leftarrow \varnothing$
14: Equivalence sets
15: $Q_i \leftarrow i, for i=1 to N$
16: While L $\neq \varnothing$ do
17: for all $v \epsilon \{L \cup Set\_s\}$ do
18: $L \leftarrow L \setminus \{v\}$
19: Set $\_s \leftarrow Set\_s \setminus \{v\}$
20: $sub\_set \leftarrow sub\_set \cup \{v\} \setminus \{p\}$.
21: Remove irrelevant feature.
22: Input CKD features to LASSO feature selection method.
23: $\widehat{\beta} = arg_\beta min \sum_{i=1}^{N} \left( y_i - \beta_0 \sum_{j=1}^{p} v_{ij}\beta_j \right)^2$
24: List the selected features from SES and LASSO

---

## Ensemble deep learning with LSTM and GRU

In phase 2, we proposed the ensemble-based deep learning approach to classify CKD. Most of the work considered individual classifiers to classify the task. Moreover, CKD

datasets are nonlinear. Using individual classifiers to classify the datasets may not produce good accuracy because of nonlinearity in the data. Therefore, this work considered the ensemble-based deep learning method to address the problems.

In Fig. 2, the proposed ensemble-learning architecture is depicted. This architecture involves training two separate deep-learning models using the same dataset and then merging their predictions to produce the final prediction result. The approach combines two distinct classifiers, with each model dedicated to a specific binary class output. We selected these two classification models with the intention that one would outperform the other in one of the output classes. This selection was based on pilot tests demonstrating the superior performance of proposed LSTM and GRU models for the majority output class 1, corresponding to chronic kidney disease. Class 0, indicating the absence of chronic kidney disease, represented the minority output.

$$f_t = \delta(w_f.[h_{t-1}, v_t] + b_f) \tag{2}$$

$$i_t = \delta(w_i.[h_{t-1}, v_t] + b_i) \tag{3}$$

$$\widetilde{C_t} = tanh(w_C.[h_{t-1}, v_t] + b_C) \tag{4}$$

$$C_t = f_t * C_t - 1 + i_t * \widetilde{C_t} \tag{5}$$

$$i_o = \delta(w_o.[h_{t-1}, v_t] + b_o) \tag{6}$$

$$h_t = o_t * tanh(C_t) \tag{7}$$

LSTM neural networks were developed using a series of interconnected neural network modules to address the problem of long-term dependency. An LSTM-based model is constructed to depict the development of CKD because of LSTM's ability to understand long-term dependencies. To forecast CKD, we have considered three layers: the pre-fully connected layer, the cells layer, and the post-fully connected layer, as described in Fig. 2. The first fully connected layer has a ReLU function and one fully connected layer; the Second Layer has one LSTM layer and a dropout wrapper; and the third layer has one completely connected layer and a softmax layer.

In an LSTM network, there are four main components: the "forget gate", "input gate", "update gate", and "output gate". The "forget gate" Eq. (2) determines whether data should be removed from the cell state. The "input gate", which consists of sigmoid and tanh layers, decides which values should be updated, and it defines Eqs. (3) and (4). The "update gate" in Eq. (5) takes the value from the "input gate" and uses it to update the

previous cell state. Finally, the "output gate" is defined in Eqs. (6) and (7) determines the value to be output from the layer. $f_t$ is a number between 0 and 1, where 0 signifies forget and 1 signifies keep; $W_f$ is the weight matrix, and $b_f$ is the bias vector. $f_t$ determines which data to be forgotten and $\widetilde{C}_t$ determines the number of cell values to be changed. The $i_o$ value in Eq. (7) determines which component of the cell state provides the output. Equation (7) yields $h_t$, the output of the components $i_o$, by multiplying the new cell state $\widetilde{C}_t$ by $o_t$ and the function *tanh* that has been chosen.

$$r_t = \delta(w_i^T(v_t) + w_i^T o(t-1) + b_r) \tag{8}$$

$$z_t = \delta(w_i^T(v_t) + w_i^T o(t-1) + b_z) \tag{9}$$

$$o_t = z_t \otimes o(t-1) + (1-z_t) \otimes o_t \tag{10}$$

## Gated recurrent unit (GRU)

In contrast to LSTM, GRU uses fewer gates in its RNN architecture. In a GRU cell unit, a single gate controls both the input and forget functions. To simplify, GRU uses a single gate for both inputs and forgets, making it less complicated than LSTM, and its equations are defined in Eqs. (8), (9) and (10). For example, when $zt = 1$, the mechanism opens the forget gate and closes the new data entry for the input gate. The new state is computed by combining the new input with the previous memory, which the reset gate decides. Where $r_t$ is the reset gate, $z_t$ is the update gate, and $o_t$ is the output gate. $\otimes$ denotes element-wise multiplication; $t$ is the time step. The window length is represented by $T$. The layer weight $w$ reflects input $v$, while b represents the output gate threshold.

## Support vector machine

An SVM learning system uses a high-dimensional feature space. SVM classifies points by distributing them to one of two separated half spaces: the pattern space or a higher-dimensional feature space. The primary goal of SVM is to find the largest margin hyperplane. The objective is to maximize the difference between positive and negative classes.

The maximum margin hyperplane is the final decision boundary. Consider that $v_i \in K^d$, $i = 1$ to N forms creates input vectors with class labels $y_i \in \{CKD - Yes, CKD - No\}$. SVM maps input vectors $v_i \in K^d$ to a high-dimensional feature space $\Phi(v_i) \in$ Hyperplane. Kernel function $kernal(v_i, v_j)$ maps $\phi(.)$. The decision boundary is defined in the Eqs. (11), (12) and (13).

$$f(v) = sgn(\sum_{i=1}^{N} y_i . \alpha_i . Kernel(v, v_i) + b) \tag{11}$$

$$Max \sum_{i=1}^{N} \alpha_i - \frac{1}{2} \sum_{i=1}^{N} \sum_{i=1}^{N} \alpha_i \alpha_j . y_i y_j . Kernel(v_i, v_j) \tag{12}$$

*subject* to

$$0 \leqslant \alpha_i \leqslant c \tag{13}$$

## Logistic regression

Logistic regression is a popular statistical analysis technique that uses predictive factors to determine the probability of an outcome. LR considered a convergence criterion to maximize likelihood functions in Eq. (14).

In this model, prob represents the probability of the result for the expected predictive variable, $k0$ is the intercept condition, $k1, k2, \ldots, kn$ are regression coefficients, and $v1, v2, \ldots, vn$ are prospective predictive variables.

$$y = (fprob) = log\left(\frac{prob}{1-prob}\right) = \alpha_0 + \alpha_1 v_1 + \alpha_2 v_2 + \alpha_n v_n. \tag{14}$$

## Decision tree

The decision tree (DT) is a machine-learning classification method based on tree structures. DT is designed to create a binary classification tree with root, internal, and leaf nodes. The root node contains input data; internal nodes represent decision function branches and leaf nodes display the output.

Training a decision tree involves two main phases: (1) constructing the classification tree and (2) pruning the tree. In the first phase, we start at the root of the classification tree and identify the input data with the highest gain ratio. Sub-nodes are then generated based on the splitting method applied to the training dataset. The second phase involves assigning a unique gain ratio value to each sub-node and using these values to establish the classification variables.

## Random Forest classifier

The Random Forest (RF) algorithm is an ensemble learning method based on decision tree learning. Ensemble learning suggests that a single classifier may not accurately classify test data because it cannot effectively differentiate between noise and patterns based on sample data. The random forest classifier trains several decision trees (n trees) using sampling with replacement from the given data sets. Each tree is trained using a random selection of three attributes. After evaluating the data, the final output is determined by the majority decision of the n trees.

## EXPERIMENTAL RESULTS

In the experimental work, we considered CKD datasets consisting of 24 features. The last variable is the class variable, which has two values: *CKD_Yes* and *CKD_No*. The dataset, initially consisting of 400 instances, is increased to 800 instances for large-scale data analysis using a technique known as oversampling (*Mamatha & Terdal, 2024*). We considered 800 samples, with 480 samples belonging to the *CKD_Yes* class and 320 samples belonging to the *CKD_No* class.

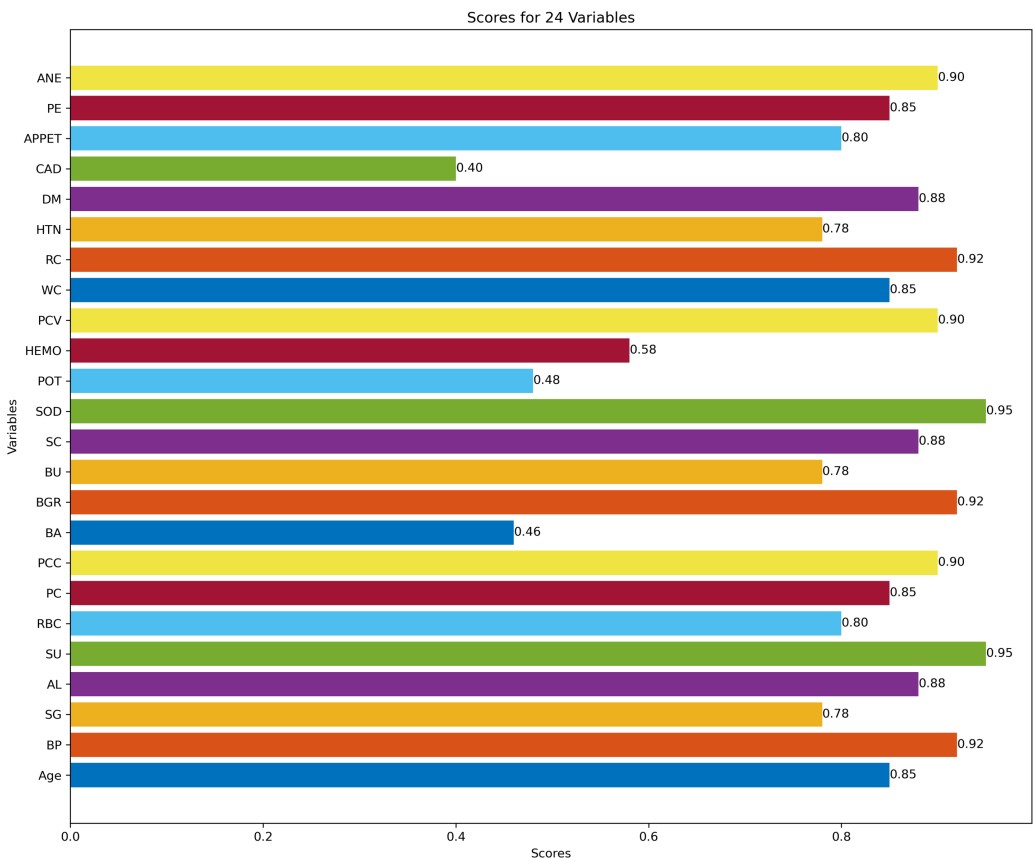

**Figure 3** **Feature score using SES feature selection method.**

Binary classification aims to identify the model that best distinguishes between the two classes. SES considers a binary classification task if the target variable is a two-level factor. We use testIndLogistic as the default conditional independence test.

Figure 3 describes feature selection using SES method. We found that these features, namely HEMO, POT, bacteria (BA), and coronary artery disease (CAD), have low scores for HEMO feature 0.58, POT feature 0.48, BA feature 0.46, and CAD feature 0.40. Figure 4 describes feature selection using the LASSO method. We found that these features, namely PE, BU, HTN, HEMO, POT, bacteria (BA), and coronary artery disease (CAD), have low scores for the PE feature 0.61, HTN feature 0.65, HEMO feature 0.64, POT feature 0.43, BA feature 0.54 and CAD feature 0.49.

The hybrid feature selection method is important because it helps identify a dataset's most relevant features or variables. This method effectively reduces the dimensionality of the data while retaining the most important information.

$$Accuracy = \frac{True\_CKDYes + True\_CKDNo}{True\_CKDYes + True\_CKDNo + False\_CKDYes + False\_CKDNo} \quad (15)$$

$$Precision = \frac{True\_CKDYes}{True\_CKDYes + False\_CKDYes} \quad (16)$$

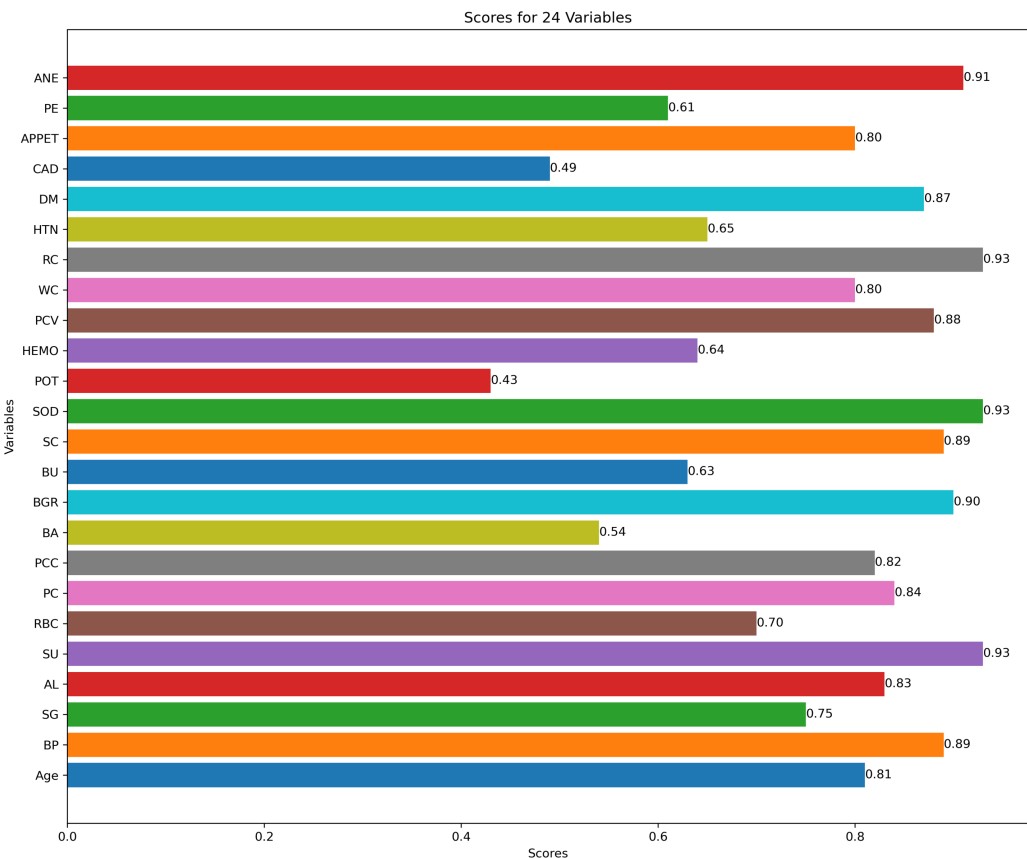

**Figure 4  Feature score using LASSO feature selection method.**

$$Recall = \frac{True\_CKDYes}{True\_CKDYes + False\_CKDNo} \qquad (17)$$

$$FScore = 2 \times \frac{Precision \times Recall}{Precision + Recall} \qquad (18)$$

In the CKD classification framework, hyperparameter tuning is vital, especially when using advanced models like LSTM networks and GRUs. Properly tuning these hyperparameters can significantly improve model performance. The learning rate, which determines the extent of weight adjustments during training, typically falls between 0.001 and 0.1. The choice of optimizer also plays a critical role; although Adam and RMSprop are favored for their adaptive learning rates and rapid convergence, experiments revealed that the stochastic gradient descent (SGD) optimizer with a learning rate of 0.02 achieved superior results. Additionally, the batch size, which indicates the number of samples processed per iteration, affects training dynamics. Smaller batch sizes (*e.g.*, 16 or 32) might introduce more noise but can accelerate convergence, while larger batch sizes (*e.g.*, 64) provide greater stability. The number of epochs, set to 50, 100, or 300, further influences

training duration and accuracy, requiring careful adjustment to optimize the model's performance. Out of all the different learning rates and optimizers we tested, we found that the SGD optimizer, when combined with a learning rate of 0.02, produced the highest level of accuracy and it is described in Figs. 5 and 6.

In the experimental work, 80% The proposed model was evaluated using accuracy, precision, recall, and F-score metric, and it is represented in Eqs. (15), (16), (17) and (18). The confusion matrix with and without feature selection is described in Figs. 7 and 8.

The experimental results of the proposed work are compared with the SVM, LR, DT, and Random Forest methods with and without feature selection methods, described in Table 3. The results of our proposed methodology demonstrate a significant performance improvement, surpassing alternative methods by a margin of 2%. The ensemble model leverages the strengths of both LSTM and GRU, allowing for improved learning and capturing of long-term dependencies in the data. By combining LSTM and GRU, the ensemble model can better generalize and adapt to a wide range of sequential data, improving performance on unseen data.

One potential direction is to explore the use of other advanced deep learning architectures, such as transformers, to improve the classification performance further. Additionally, conducting a comparative analysis with other state-of-the-art feature selection and classification methods could provide valuable insights into the strengths and weaknesses of the proposed approach. Furthermore, integrating multi-modal data sources, such as genetic information or histopathology images, could enhance the model's predictive capabilities. Finally, conducting a clinical validation study to assess the real-world performance of the proposed model would be an essential step toward its practical deployment in healthcare.

## CONCLUSION

The proposed research on CKD classification is divided into two distinct phases, each aimed at improving diagnostic accuracy through proposed methodologies. In the first phase, a hybrid feature selection method was employed to identify the most predictive features for CKD classification. This method combines the strengths of two approaches: the Bayesian network-based SES method, which identifies relevant predictors through causal analysis, and LASSO, which performs regression to select features with the highest correlations while shrinking the coefficients of less significant features to zero. This hybrid approach effectively reduces dataset dimensionality and focuses on the most impactful features, minimizing noise and redundancy. The second phase of the paper introduced an ensemble-based deep learning model, integrating LSTM and GRU networks. This hybrid model leverages the sequential data processing capabilities of LSTM, which captures long-term dependencies through its gating mechanisms, and the computational efficiency of GRU, which simplifies the LSTM architecture while maintaining strong performance. The ensemble approach was designed to capitalize on the strengths of both networks, enhancing classification accuracy for CKD. Various learning rates (0.01, 0.02, 0.03, and 0.04) and optimizers (Adam, RMSprop, and SGD) were tested for experimental validation. The results showed that a learning rate 0.02 combined with the SGD optimizer yielded the highest classification

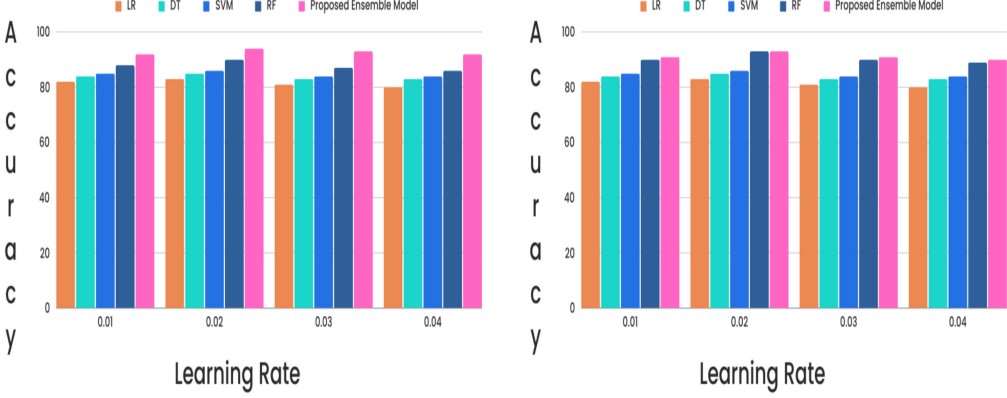

(a) Adam optimization method          (b) RMSprop optimization method

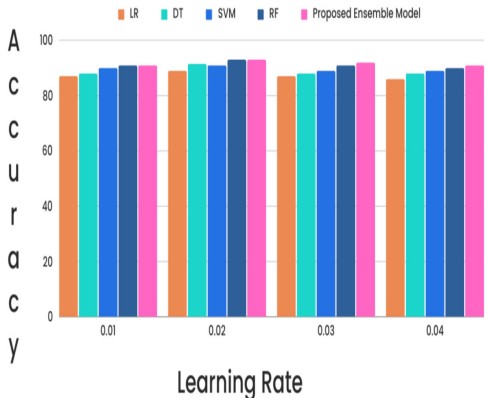

(c) SGD optimization method

**Figure 5** (A–C) Hybrid feature selection method.

accuracy. This combination outperformed other settings, highlighting its effectiveness for the proposed deep learning model. Comparative analysis was conducted with individual classifiers such as DT, RF, LR, and SVM. The proposed hybrid feature selection method and the LSTM-GRU ensemble model demonstrated a 2% improvement in classification accuracy over these traditional methods. The feature analysis revealed that certain features, including HEMO, POT, bacteria, and coronary artery disease, were non-contributory to

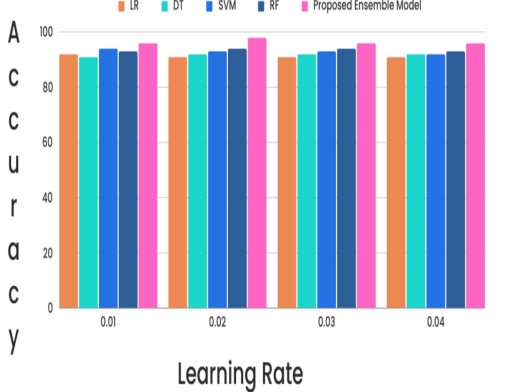

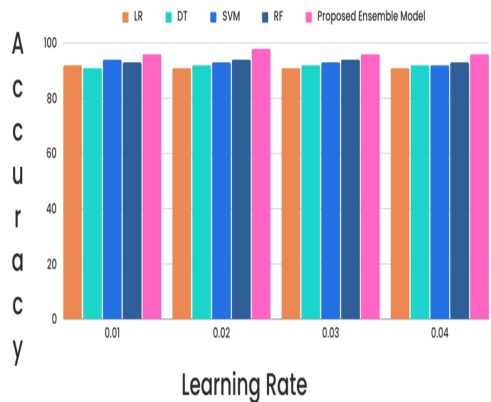

**(a)** Adam optimization method

**(b)** RMSprop optimization method

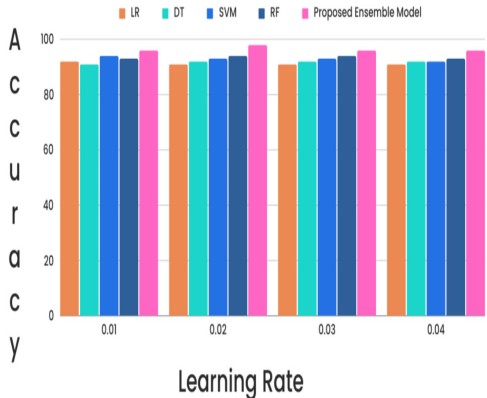

**(c)** SGD optimization method

**Figure 6** **(A–C) Without feature selection method.**

the classification tasks, guiding future refinement efforts. Future research could further optimize deep learning model hyperparameters, such as the number of layers, hidden units, learning rates, and dropout rates. Automated techniques like Bayesian optimization or genetic algorithms could efficiently explore the hyperparameter space and identify the optimal configurations for enhancing model performance. This future work will aim to

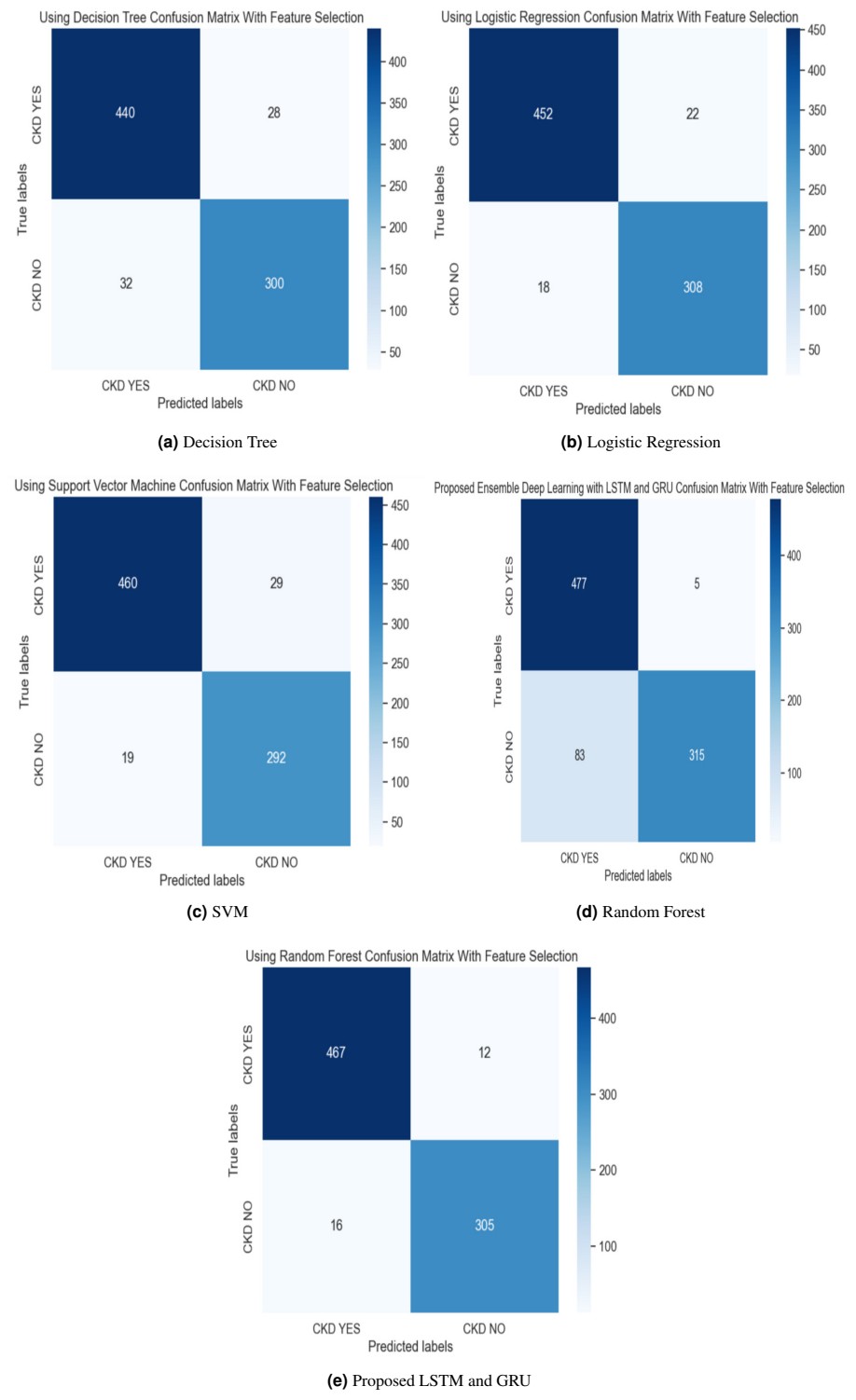

**Figure 7** (A–E) Confusion matrix using hybrid feature selection method.

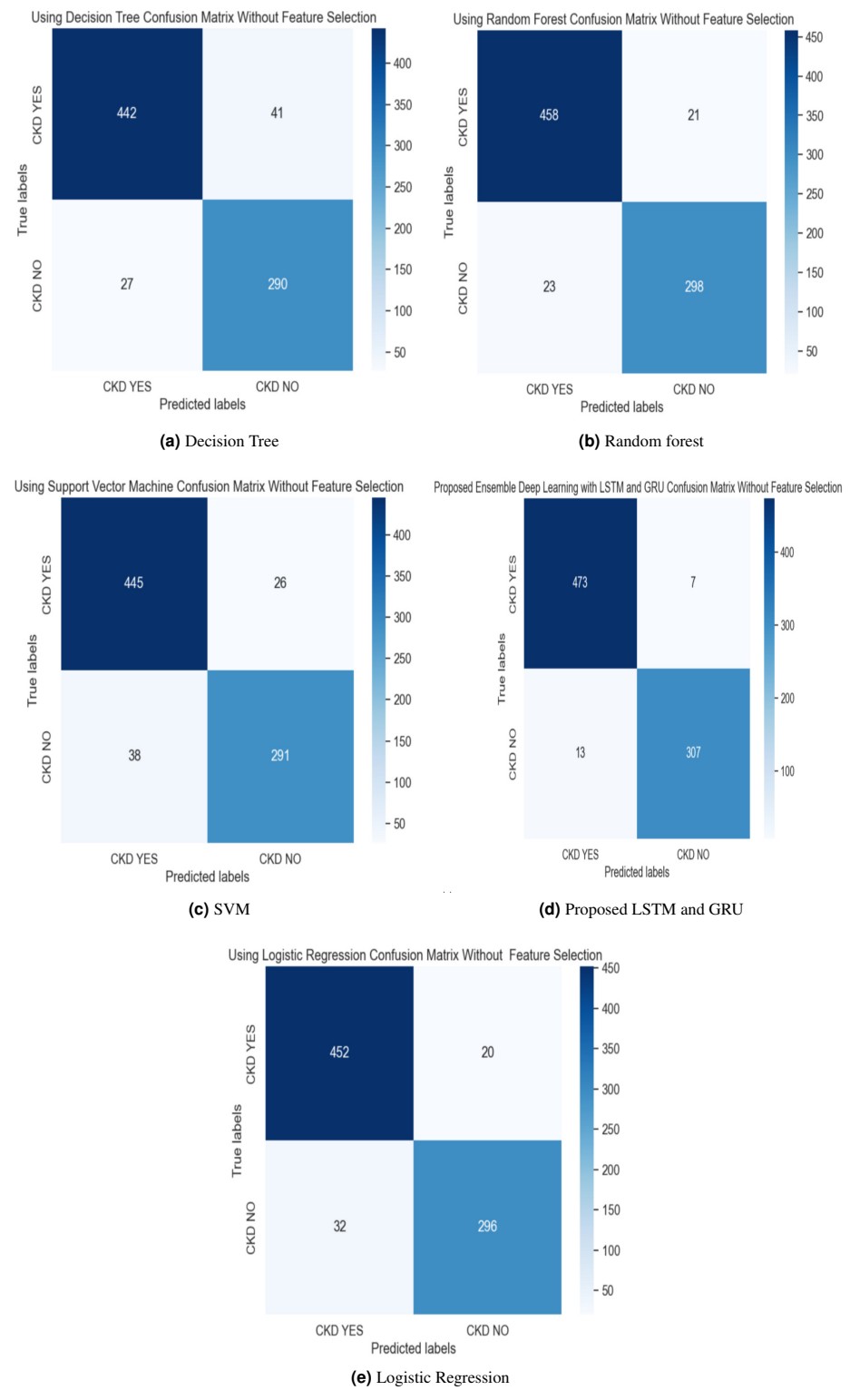

**Figure 8** (A–E) Confusion matrix without feature selection method.

**Table 3  Results comparison.**

| Model | Feature selection | Accuracy (%) | Precision (%) | Recall (%) | F-score |
|---|---|---|---|---|---|
| Support vector machine | No | 0.92 | 0.92 | 0.94 | 0.93 |
| Logistic regression | No | 0.94 | 0.93 | 0.96 | 0.95 |
| Decision tree | No | 0.92 | 0.94 | 0.93 | 0.92 |
| Random forest | No | 0.95 | 0.95 | 0.96 | 0.96 |
| **Proposed ensemble deep learning** | No | 0.98 | 0.97 | 0.99 | 0.98 |
| Support vector machine | Proposed hybrid feature selection | 0.94 | 0.96 | 0.94 | 0.95 |
| Logistic regression | Proposed hybrid feature selection | 0.95 | 0.96 | 0.95 | 0.96 |
| Random forest | Proposed hybrid feature selection | 0.97 | 0.97 | 0.97 | 0.97 |
| **Proposed ensemble deep learning** | **Proposed hybrid feature selection** | 0.99 | 0.85 | 0.99 | 0.92 |

refine the classification process, potentially leading to even greater improvements in CKD diagnostic accuracy.

**Abbreviation Meaning**

| | |
|---|---|
| **ANN** | Artificial Neural Networks |
| **CKD** | Chronic Kidney Disease |
| **CNN** | Convolutional Neural Network |
| **HMANN** | Heterogeneous Modified Artificial Neural Network |
| **KNN** | K-Nearest Neighbour |
| **KFUH** | King Fahd University Hospital |
| **LR** | Logistic regression |
| **NB** | Naive Bayes |
| **RBF** | Radial Basis Functions |
| **RF** | Random Forest |
| **ROC Curve** | Receiver Operating Characteristic curve |
| **RFE** | Recursive Feature Elimination |
| **RNN** | Recurrent Neural Network |
| **SVM** | Support Vector Machine |
| **UCI** | University of California Irvine |

# ACKNOWLEDGEMENTS

The authors acknowledge Editor-in-chief of this journal for the constant encouragement to finalize the article. Also, the authors are grateful for the comments and suggestions by the referee. Their comments and suggestions greatly improved the article.

## Funding

The authors received no funding for this work.

## Competing Interests

The authors declare there are no competing interests.

## Author Contributions

- Yogesh N conceived and designed the experiments, performed the experiments, analyzed the data, performed the computation work, prepared figures and/or tables, authored or reviewed drafts of the article, and approved the final draft.
- Purohit Shrinivasacharya conceived and designed the experiments, performed the computation work, prepared figures and/or tables, authored or reviewed drafts of the article, and approved the final draft.
- Nagaraj Naik conceived and designed the experiments, performed the experiments, analyzed the data, performed the computation work, prepared figures and/or tables, authored or reviewed drafts of the article, and approved the final draft.

## Data Availability

The Chronic Kidney Disease dataset is available at: Rubini, L., Soundarapandian, P., & Eswaran, P. (2015). Chronic Kidney Disease [Dataset]. UCI Machine Learning Repository. https://doi.org/10.24432/C5G020.

## Supplemental Information

Supplemental information for this article can be found online at http://dx.doi.org/10.7717/peerj-cs.2467#supplemental-information.

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
