# Peer review of "Novel statistically equivalent signature-based hybrid feature selection and ensemble deep learning LSTM and GRU for chronic kidney disease classification"

_PeerJ Computer Science, doi:10.7717/peerj-cs.2467_

## Round 0.1 · original submission · Major Revisions

Dear Authors,
Thank you for submitting your manuscript titled "Deep Learning, Ensemble, and Classification Methods for CKD Prediction" to our journal. We have received feedback from three reviewers, and after careful consideration, we have decided that the manuscript requires major revisions before it can be considered for publication.
Reviewer 1 has noted several critical points that need addressing:
1. The significance and goal of your research should be explicitly stated, and the necessity of your proposed effort needs to be justified, especially given the existing CKD prediction techniques.
2. The research gap is not clearly defined, and evidence for the necessity of your task should be provided.
3. Revisions are needed in the abstract and conclusion sections to better reflect the completed work.
4. The use of ensemble techniques needs to be justified, particularly concerning the time and cost relative to the improvement in accuracy.
5. The literature review should include an analysis of deep learning and ensemble studies.
6. The choice of techniques used in your study requires clarification.
7. Additional explanations are required for the attribute selection process, data set description, and model representation.
Reviewer 2 has recommended rejection but provided constructive feedback that can be addressed to improve your manuscript:
1. Justification is needed for the use of LSTM in the context of your problem statement.
2. The data used in your study is outdated (2005); more recent data should be considered.
3. Validation from nephrologists is necessary to ensure the selected parameters' validity.
4. A discussion on hyperparameter tuning for GRU and LSTM is required.
5. The hybrid feature selection part needs to be demonstrated in the code.
Reviewer 3 has suggested minor revisions but also pointed out important areas that need attention:
1. Clarification is needed regarding the use of SES and LASSO models, as the current code does not reflect their usage.
2. Minor corrections in sentences on line 19 and 54 should be made.
To improve your manuscript, we recommend addressing the points raised by all three reviewers thoroughly. Please ensure that you:
• Clearly state the significance and goals of your research.
• Define the research gap and provide evidence for the necessity of your work.
• Justify the use of ensemble techniques and LSTM, considering the context and requirements of your problem statement.
• Update the data used and validate the results with relevant experts.
• Include detailed discussions on hyperparameter tuning and feature selection methods.
• Revise the abstract, conclusion, and any minor language issues as suggested.
We believe that addressing these points will significantly strengthen your manuscript. We look forward to receiving your revised submission.
Best regards,

·

Basic reporting

1. The research work meets its goals and makes a valuable contribution.
2. The suggested work focused on deep learning, ensemble, and classification methods for CKD prediction.
3. The significance and goal of the suggested research must be stated by the authors. What is the need for the proposed effort, given the abundance of improved CKD prediction techniques now available?
4. The research gap is not clearly defined; hence the authors must provide evidence for the necessity of the suggested task.
5. The abstract and conclusion sections need to be revised in light of the completed work.
6. Why is the use of ensemble techniques necessary?
7. For a small improvement in the outcomes, particularly accuracy when employing several strategies What's the estimated time and expense? Justify
8. The analysis of deep learning and ensemble studies has to be added to the literature review.
9. The authors need to explain why they choose to use particular techniques. Why did the authors select the current research techniques?

Experimental design

1. The introduction and abstract sections need to include an explanation of the proposed research work
2. A general grammatical check is necessary
3. The writers stated that choosing a characteristic is a crucial responsibility. An explanation of the attribute selection and its outcomes is required.
4. A description of the data set is necessary.
5. A model representation for the contributions is suggested

Validity of the findings

1. The unit representation in the tables is necessary.
2. Justify the SES and LESSO characteristics in the method.
3. There is uncertainty in every figure. The numbers ought to be obvious.
4. Explain the significance of group work and in-depth education.

Reviewer 2 ·

Basic reporting

Clear and unambiguous, professional English used throughout.

Experimental design

Introduction can be modified with the help of statistics to understand the need of CKD.
The purpose of using LSTM is for tasks which are time bound/if there is any need for time series analysis. How can you justify the use of LSTM for a problem statement like this.

Validity of the findings

The data on which the work has been done is a very old data(2005).Authors need to look for recent data which can have new features as well.
The results should be validated from any Nephrologists, to ensure that the parameters selected by the algorithm are valid.
There has to be a discussion on the hyperparameter tuning part for GRU and LSTM. Since these networks will not perform well without the same.
The code does not demonstrate the hybrid feature selection part anywhere. Authors need to validate the same.

Additional comments

The proposed architecture is good. But validity of results needs to be verified once by the authors.

Reviewer 3 ·

Basic reporting

Article is well presented with appropriate introduction and background. Literature presented is relevant.
contribution of the work is well written and conforms to the experimental work undertaken.

Experimental design

Dataset source is mentioned.
architecture framework of the proposed work is well depicted.
investigation of evaluation metrics is well documented.
Feature selection methods SES and LASSO used is presented well along with the algorithm. similarly LSTM ,GRU model study and implementation is discussed along with code in review materials. however code does not reflect the use of SES and LASSO models. Clarification about the same is expected.
Evaluation study conducted is also satisfactory.

Validity of the findings

Evaluation study is satisfactory.
conclusion is clearly stated with future scope.

Additional comments

overall paper is well presented.
Authors may look at minor corrections in sentences in Line number 19 and 54.
Line 19 "Most feature "selection (FS) classifiers need better only for strongly correlated data.
Line 54 " Few studies have investigated effective techniques adapted to the classification task.."

---

## Round 0.2 · Minor Revisions

Dear Authors,

Pay attention to the final minot suggestions made by reviewer 1.

Thank you so much.

·

Basic reporting

The paper is rewritten carefully, and the revisions have been done properly

Experimental design

1. An overall model diagram for the proposed method is suggested
2. The metrics and units are required in all tables

Validity of the findings

1. The work done in CKD classification is satisfactory
2. The results obtained in the experimentation are satisfactory
3. The Deep Learning based classification techniques are given acceptable results

Additional comments

The recent studies should be added to the review of literature

Reviewer 3 ·

Basic reporting

no comment

Experimental design

based on the query raised relevant code for saso,ses is presented and satisfactory

Validity of the findings

satisfactory

Additional comments

corrections made as per the query and comments given. Overall the work is satisfactory.

---

## Round 0.3 · accepted · Accept

All the suggestions have been addressed. Thank you very much.